# A Case of Sotos Syndrome in a Preterm Infant with Severe Bronchopulmonary Dysplasia and Congenital Heart Disease

**DOI:** 10.3390/children10071111

**Published:** 2023-06-26

**Authors:** Dan-Fang Lu, Xiao-Mei Tong, Yun-Feng Liu

**Affiliations:** Department of Pediatrics, Peking University Third Hospital, Beijing 100191, China; ludanfang2021@163.com (D.-F.L.); tongxm2007@126.com (X.-M.T.)

**Keywords:** Sotos syndrome, preterm infant, bronchopulmonary dysplasia, congenital heart disease

## Abstract

Sotos syndrome is an autosomal dominant genetic disorder caused by mutations in the NSD1 gene. In this study, we report a case of Sotos syndrome in a preterm infant. The main clinical manifestations were severe bronchopulmonary dysplasia, congenital heart disease, difficulty feeding, and characteristic facial appearance. The gene mutation was located at 177251854 on chromosome 5, and identified as a shear mutation, c.4765+1 G > A, which is a new mutation. The patient recovered well after symptomatic treatment. To the best of our knowledge, this is the first case of a preterm infant in whom a novel c.4765+1 G > A mutation in the NSD1 gene was identified. When premature infants present with abnormally severe bronchopulmonary dysplasia, feeding difficulties, and other congenital anomalies, Sotos syndrome should be considered.

## 1. Introduction

Sotos syndrome (SoS) is a childhood overgrowth condition, first described in 1964 by Sotos et al. [1]. It is characterized by three cardinal features: a distinctive facial appearance (including a broad and prominent forehead with a dolichocephalic head shape, sparse frontotemporal hair, down-slanting palpebral fissures, a long and narrow face, and a long chin), learning disability, and overgrowth. Neonates with SoS may also have jaundice, hypotonia, and poor feeding [2]. To date, and to the extent of our knowledge, there are no reports in the literature on SoS with severe bronchopulmonary dysplasia (BPD) and congenital heart disease (CHD) in preterm infants.

## 2. Case Report

### 2.1. Birth History

A newborn boy was born via cesarean section at 31 weeks and 4 days of gestation due to intrauterine distress. The newborn had a limp tone and poor respiratory effort on delivery, with a birth weight of 1820 g (64th percentile), length of 42 cm (59th percentile), and head circumference of 29 cm (50th percentile). He was transferred to the neonatal intensive care unit at our center due to respiratory distress. Apgar scores at 1 min, 5 min, and 10 min were 7, 10, and 10, respectively. The prenatal ultrasound and magnetic resonance imaging (MRI) showed ventricular widening and polyhydramnios (Figure 1). A prenatal diagnosis of amniocentesis showed no karyotype or chromosomal abnormalities. The 32-year-old mother underwent prenatal in vitro fertilization and embryo transfer for ovarian cysts and developed hypertension during pregnancy.

### 2.2. Clinical Manifestation

After the infant was admitted to the neonatal intensive care unit, respiratory distress worsened and blood gas analysis indicated severe respiratory failure; thus, pulmonary surfactant and non-invasive ventilator support were provided. At 7 h after birth, the newborn was intubated because of increased dyspnea, hypotonia, decreased responsiveness, and decreased blood pressure. Sepsis was suspected, and he was immediately treated with antibiotics. However, his complete blood count was normal and blood culture was negative. Echocardiography (ECHO) suggested the presence of a large patent ductus arteriosus (PDA) and pulmonary hypertension; thus, he was treated with dopamine, dobutamine, norepinephrine, sildenafil, and ibuprofen. The infant’s respiratory condition improved 6 days after birth and the endotracheal tube was removed. However, after the endotracheal tube was removed and changed to non-invasive ventilation, the baby soon redeveloped respiratory distress. He underwent two failed extubation attempts, until 21 days after birth. Over this time, ECHO confirmed the presence of an atrial septal defect, and infection was excluded. Subsequently, he experienced two episodes of pneumonia. At 36 weeks postmenstrual age (PMA), the patient was still dependent on non-invasive intermittent positive pressure ventilation; his FIO2 was 0.3, and he was diagnosed with severe, grade III BPD. Upon weight gain, we found changes in his appearance, including forehead protrusion, pointed chin, widening eye distance, wide-set down-slanting eyes, high arched palate, skin relaxation, and pigmentation (Figure 2). The infant did not begin sucking until 40 weeks PMA and had a generally weaker suck. On day 91 after birth (44 weeks and 4 days PMA), the infant was discharged from the hospital, and further oxygen therapy and partial nasal feeding were performed at home.

### 2.3. Accessory Examination

No ureaplasma urealyticum, virus, or bacteria were found in the etiological examination of the sputum. Bedside ECHO revealed an atrial septal defect (ASD) with a diameter of 5 mm and a left-to-right shunt; no ductus arteriosus flow was detected at 6 days after birth. Pulmonary edema and exudative inflammation were noted in bilateral lung fields on lung ultrasound. Fifty-one days after birth, pulmonary computed tomography (CT) suggested a diffuse ground-glass appearance in both lungs (Figure 3). Cranial MRI at 91 days after birth showed bilateral ventricular widening and corpus callosum hypoplasia (Figure 4). Due to the infant’s characteristic appearance, repeated weaning failures, and congenital heart disease, we considered the possibility of a congenital genetic disease. Whole-exome sequencing (WES) revealed SoS type I. However, sequencing of both parents of the proband yielded full-length wild-type transcripts, suggesting that the disease was a genetic mutation (Figure 5).

### 2.4. Follow-Up

Post-discharge follow-up was performed one month after discharge, during which the patient underwent ASD repair and PDA ligation. The infant was fed normally at 4 months PMA. He is now 10 months PMA, has overgrowth, delayed language, and motor development, and still requires overnight oxygen due to left bronchial stenosis.

## 3. Discussion 

SoS occurs in approximately 1:10,000–1:15,000 live births, with an autosomal dominant pattern of inheritance. In approximately 90% of patients, SoS is associated with mutations in *NSD1*, protein insufficiency, and a 5q35 microdeletion [3]. Abnormalities in this NSD1 protein lead to uncontrolled overgrowth [4,5]. In this case, the gene mutation was located at 177,251,854 on chromosome 5; a shear mutation c.4765+1G > A. No mutation site was found in either parent, and to the best of our knowledge, this novel *NSD1* gene mutation has not yet been described in the literature, nor published in mutation databases. Not many premature infants have been diagnosed with SoS type 1 during the neonatal period.

The phenotype of NSD1-positive individuals is extremely variable. The three cardinal features of SoS were present in 90% of affected individuals; however, many other features were also variably present [6]. Neonates may also have jaundice (approximately 65%), hypotonia (approximately 75%), and poor feeding (approximately 70%), as well as cardiac anomalies (including PDA, ASD, and ventricular septal defects) [2] and/or cranial CT/MRI abnormalities (including ventricular dilatation, hypoplasia, agenesis of the corpus callosum, or mega cisterna magna) [7]. Other uncommon features include neonatal hypoglycemia, hyperinsulinemia, skin pigmentation, hypothyroidism, and inguinal hernia [6,8,9,10,11]. In this case, the clinical manifestations in the infant were consistent with existing reports found in the literature. However, the ASD in this child increased with age and the PDA reopened after closing, which required surgical intervention within 6 months, making it a relatively serious cardiac dysplasia. In addition, the lung lesions in this child were very prominent, and no similar cases have been reported in existing studies. Severe lung lesions may be associated with abnormal expression of the protein encoded by the NSD1 gene in the lungs [12]. In the absence of oral intervention, preterm infants can achieve total oral feeding at approximately 34 weeks PMA [13]. However, this child was not released from a feeding tube until 4 months PMA, which may be related to hypotonia and gastroesophageal reflux, caused by SoS. This is consistent with the literature reporting that feeding difficulties can recover with age [2,14]. Thus, in addition to his characteristic appearance, severe congenital heart disease, severe lung lesions, and feeding difficulties suggested that the infant had SoS.

The diagnosis of SoS requires a comprehensive analysis of the clinical phenotype, and a final genetic diagnosis is essential. For premature infants, SoS can be difficult to identify based on the three primary characteristics. Some infants do not show significant cosmetic abnormalities until 1–2 years after birth [15]. In this case, the appearance was not detected until 36 weeks PMA. Overgrowth and stunting did not occur until 10 months PMA. Therefore, in the neonatal period, in addition to the three major features, we need to pay attention to other clinical manifestations, such as jaundice, hypotonia, feeding difficulties, skin pigmentation, ventricular dilatation, recurrent lung infections, and CHD. For premature infants, severe BPD inconsistent with gestational age should also be considered. The diagnosis of SoS was confirmed via WES. A prenatal diagnosis of SoS may be beneficial in terms of delivery plans and clinical intervention. Fetuses with SoS may prenatally manifest with increased nuchal translucency (NT), an increased risk of Down syndrome on maternal serum screening, macrocephaly, polyhydramnios, fetal overgrowth, renal abnormalities, and central nervous system abnormalities [16,17]. The reported neuroimaging findings included enlargement of the lateral ventricles, trigones, and occipital horns, corpus callosum hypoplasia, persistence of cavum septum pellucidum, cavum vergae, and cavum velum interpositum, enlarged cisterna magna, heterotopias, macrocerebellum, and periventricular leukomalacia [18,19]. A prenatal ultrasound and non-invasive prenatal screening for a panel of dominant single-gene disorders (NIPT-SGD) of amniotic fluid aid in prenatal diagnosis [20,21]. Fetal ECHO screening should be performed as early as possible when a prenatal ultrasound indicates increased NT, polyhydramnios, and any other abnormalities [22].

A summary of the prenatal and postnatal clinical features of SoS are presented in Table 1, below.

There is no standard course of treatment for SoS, as treatment is solely symptomatic [2]. In this case, the patient received targeted treatment for pulmonary dysplasia and repeated pulmonary infections, along with a focus on nutrition. After discharge, he was transferred to the children’s cardiac surgery department for further cardiac surgery, and then transferred to the children’s health care department for follow-up on his growth and development.

SoS is not a life-threatening disorder, and patients may have a normal life expectancy. The initial abnormalities characteristic of SoS usually resolve as the growth rate normalizes, after the first few years of life [14]. Thus, it is expected that the child will lead a normal life following the performed interventions.

## 4. Conclusions

SoS is an autosomal dominant genetic disease with three clinical manifestations: a characteristic appearance, cognitive impairment, and overgrowth. The report of this case emphasizes that, in addition to the above manifestations, congenital cardiac conditions, severe BPD, and repeated pulmonary infections are likely to be prominent in premature infants. WES can be used to confirm the diagnosis. Prenatal genetic diagnosis facilitates clinical decision making.

## Figures and Tables

**Figure 1 children-10-01111-f001:**
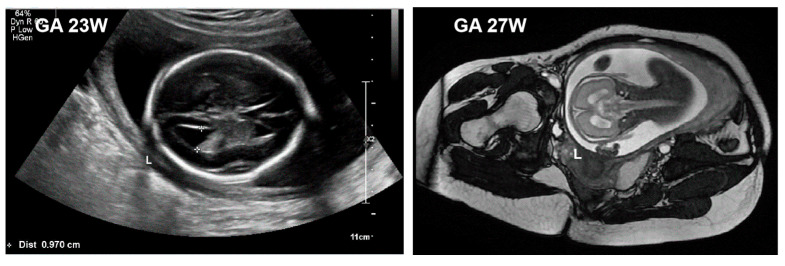
Both fetal cranial ultrasound at 23 weeks and fetal MRI at 27 weeks of gestation indicated left ventricle widening (10 mm).

**Figure 2 children-10-01111-f002:**
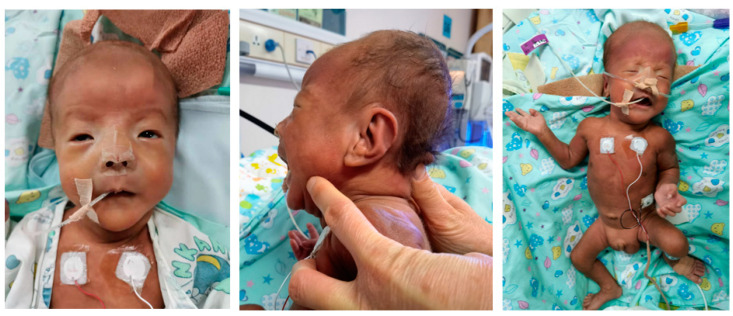
Facial gestalt for the child with Sotos syndrome (forehead protrusion, pointed chin, widening eye distance, wide-set down-slanting eyes, high arched palate, skin relaxation, and pigmentation).

**Figure 3 children-10-01111-f003:**
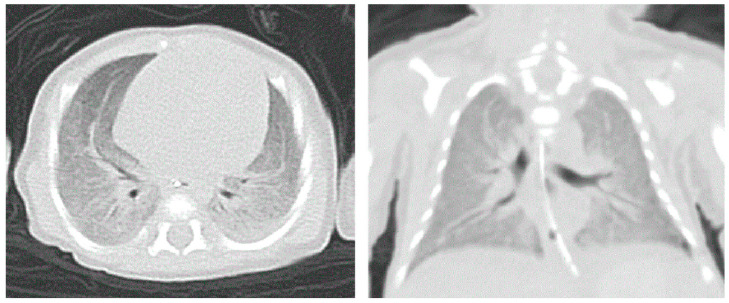
Pulmonary CT at 51 days suggested a diffuse ground-glass appearance in both lungs.

**Figure 4 children-10-01111-f004:**
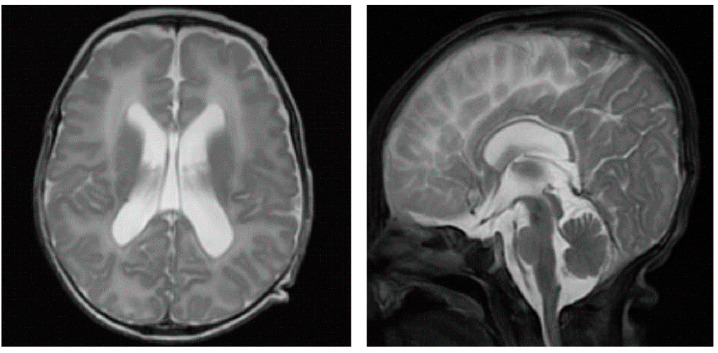
Cranial MRI at 91 days after life suggests bilateral ventricular widening and corpus callosum hypoplasia.

**Figure 5 children-10-01111-f005:**
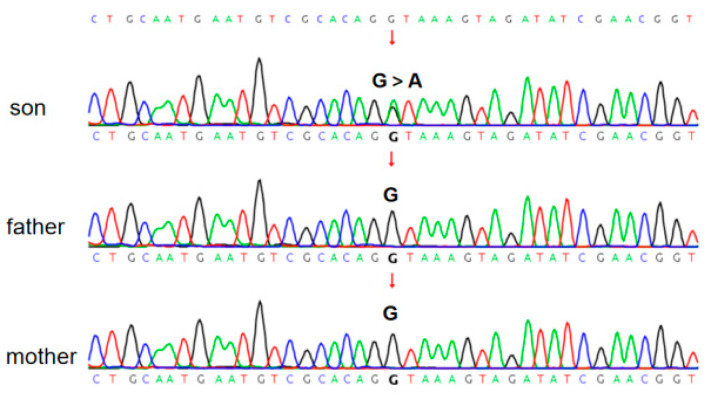
WES of the parents and the patient. The c.4765+1 G > A locus mutations were detected in the genetic sequence of the son only. No mutation site was found in the parents. The mutation sites are indicated with the arrows.

**Table 1 children-10-01111-t001:** The prenatal and postnatal clinical features of SoS.

Prenatal Features	Postnatal Features
Increased NT	Characteristic appearance
Abnormal maternal serum screening results	Hypotonia, jaundice, poor feeding
Macrocephaly, fetal overgrowth	Macrocephaly, overgrowth
Polyhydramnios	Severe lung lesions
Central nervous system abnormalities	Ventricular dilatation, hypoplasia, agenesis of the corpus callosum, mega cisterna magna
Renal abnormalities	Cardiac anomalies (including PDA, ASD, and ventricular septal defect)
	Uncommon: Neonatal hypoglycemia, hyperinsulinemia, skin pigmentation, hypothyroidism, and inguinal hernia

SoS—Sotos syndrome; NT—nuchal translucency; PDA—patent ductus arteriosus; ASD—atrial septal defect.

## Data Availability

No new data were created or analyzed in this study. Data sharing is not applicable to this article.

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
