# Peer review of "A Case of Sotos Syndrome in a Preterm Infant with Severe Bronchopulmonary Dysplasia and Congenital Heart Disease"

_children, 2023, doi:10.3390/children10071111_

Round 1

Reviewer 1 Report

The authors presenting a Sotos syndrome in a premature infant. the case report is well written and here are two advices for the authors that might improve the quality of their manuscript:
1- They should include the gestational weeks.
2- It will be helpful if they can include MRI images for brain and chest.

Author Response

1- They should include the gestational weeks.
Thank you for your comments. gestational weeks has been added to the article and marked in yellow.
2- It will be helpful if they can include MRI images for brain and chest.
Thank you for your comments. The case report added the images of cranial MRI and pulmonary CT during the infant's hospitalization(figure4 and figure5), it is regrettable that the infant did not have a pulmonary MRI.

Reviewer 2 Report

THE MANUSCRIPT DESCRIBES AA ANOMALY NOT VERY RARE

IN PUBMED THERE ARE SIMILAR CASES

THE ARTICLE NEEDS ADDITIONAL INFORMATION     AND DETAILS IN ALL PARAGRAPHS TO BE IMPROVED AND RESUBMITTED

THE CASE REPORT SHOULD BE DIVIDED IN MORE SUBCHAPTERS

Author Response

1、THE MANUSCRIPT DESCRIBES AN ANOMALY NOT VERY RARE IN PUBMED THERE ARE SIMILAR CASES
Thank you for your comments. Although some studies have reported Sotos syndrome, Sotos syndrome detected during the hospitalization of the preterm infants is uncommon, and this case report will help NICU doctors to find some similar cases.
2、THE ARTICLE NEEDS ADDITIONAL INFORMATION AND DETAILS IN ALL PARAGRAPHS TO BE IMPROVED AND RESUBMITTED
Thanks for your comments. You can ask some details, I hope to better correct.
3、THE CASE REPORT SHOULD BE DIVIDED IN MORE SUBCHAPTERS
Thanks for your comments.The case report section has been written in subsections as requested and marked in yellow.

Reviewer 3 Report

This is a well written case report regarding Soto syndrome in a preterm infant. However, may preterm infants present with severe BPD and congenital cardiac conditions such as ASD/PDA, therefore I think more focus should be made on the other aspects that made you consider Soto syndrome in this patient. 

Specific comments:

-Abstract, page 1, line 10: Change from special appearance to something that more describes that it is a characteristic facial appearance.

-Case report, page 2, line 27: State the patients PMA at time of delivery for clarification of how premature the infant was at delivery.

-Case report, page 2, line 45: edit grammar of functional closure statement.

-Case report, page 2, line 46: Describe why you decided to obtain WES. Was it because of the characteristic physical appearance that you had concern for Soto at this point, was it that the clinical picture was unclear, etc?

-Discussion, page 3, line 92: Again, elucidate what made you think this might be Soto.

-Conclusion, page 3, line 111: I would add congenital cardiac conditions, because as your title suggests that was a main component that made you recognize this might be Soto.

I only recognize minor revisions in English language needed.

Author Response

Specific comments:
1、Abstract, page 1, line 10: Change from special appearance to something that more describes that it is a characteristic facial appearance.
Thanks for your comments.The article has been modified as required and marked in yellow as indicated.
2、Case report, page 2, line 27: State the patients PMA at time of delivery for clarification of how premature the infant was at delivery.
Thanks for your comments.The patients PMA at time of delivery has been modified as required and marked in yellow as indicated.
3、Case report, page 2, line 45: edit grammar of functional closure statement.
Thanks for your comments.The article has been modified as required and marked in yellow as indicated.
4、Case report, page 2, line 46: Describe why you decided to obtain WES. Was it because of the characteristic physical appearance that you had concern for Soto at this point, was it that the clinical picture was unclear, etc?
Thanks for your comments.The reasons for the WES inspection have been stated in the article and marked in yellow.
5、Discussion, page 3, line 92: Again, elucidate what made you think this might be Soto.
Thanks for your comments.The article has been modified as required and marked in yellow as indicated.
6、Conclusion, page 3, line 111: I would add congenital cardiac conditions, because as your title suggests that was a main component that made you recognize this might be Soto.
Thanks for your comments.I have added it to the discussion and marked it yellow.

Reviewer 4 Report

Dear authors, 

congratulations I've loved to read your paper

Genetic disorders are rare, it is crucial to publish every new mutation leading to a diagnosis in order to help further fetal medicine specialists and pediatrician in correctly understand the syndrome they are facing

i have minor revisions to suggest

1) please within the case presentation add images of the two prenatal features of abnormalities

2) please within the discussion please briefly describe prenatal findings typical of Sos

3) please within the discussion highlight the crucial importance to perform a fetal echocardiography in front of ANY abnormality detected on ultrasound (read PMID: 36786908 

4) please add a table with a summary of prenatal and postnatal features of Sos

best regards

Author Response

1、please within the case presentation add images of the two prenatal features of abnormalities
Thanks for your comments. I have add two images of features of abnormalities.
2、please within the discussion please briefly describe prenatal findings typical of Sos SOS
Thanks for your comments. I have describe prenatal findings typical of Sos in the discussion and marked it yellow.
3、please within the discussion highlight the crucial importance to perform a fetal echocardiography in front of ANY abnormality detected on ultrasound (read PMID: 36786908 )
Thank you for your review. The importance of fetal echocardiography has been highlighted in the discussion and marked it yellow.
4、please add a table with a summary of prenatal and postnatal features of Sos
Thank you for your review. A summary of prenatal and postnatal features of Sos has been added to the discussion and marked it yellow.